# Advances of Genomic Medicine in Psoriatic Arthritis

**DOI:** 10.3390/jpm12010035

**Published:** 2022-01-03

**Authors:** Carlos M. Laborde, Leyre Larzabal, Álvaro González-Cantero, Patricia Castro-Santos, Roberto Díaz-Peña

**Affiliations:** 1Dreamgenics, 33010 Oviedo, Spain; carlos.martinez@dreamgenics.com (C.M.L.); leyre.larzabal@dreamgenics.com (L.L.); 2Department of Dermatology, Hospital Universitario Ramon y Cajal, 28034 Madrid, Spain; alvarogc261893@hotmail.com; 3Faculty of Medicine, Universidad Francisco de Vitoria, Ctra. Pozuelo-Majadahonda, 28223 Pozuelo de Alarcón, 28034 Madrid, Spain; 4Immunology, Centro de Investigaciones Biomédicas (CINBIO), Universidad de Vigo, 36310 Vigo, Spain; patricassan@gmail.com; 5Faculty of Health Sciences, Universidad Autónoma de Chile, Talca 3460000, Chile

**Keywords:** psoriatic arthritis, spondyloarthropathies, personalized medicine, genomics, genome-wide association studies, transcriptomics, epigenomics

## Abstract

Psoriatic arthritis (PsA) is a common type of inflammatory arthritis found in up to 40% of patients with psoriasis. Although early diagnosis is important for reducing the risk of irreversible structural damage, there are no adequate screening tools for this purpose, and there are no clear markers of predisposition to the disease. Much evidence indicates that PsA disorder is complex and heterogeneous, where genetic and environmental factors converge to trigger inflammatory events and the development of the disease. Nevertheless, the etiologic events that underlie PsA are complex and not completely understood. In this review, we describe the existing data in PsA in order to highlight the need for further research in this disease to progress in the knowledge of its pathobiology and to obtain early diagnosis tools for these patients.

## 1. Introduction

Psoriatic arthritis (PsA), or the broader term psoriatic disease, refers to an inflammatory disorder that affects numerous organs, including the skin and joints. It is also associated with extra-articular manifestations and multiple comorbidities [1]. In 1964, PsA was recognized as a separate disease by the American Rheumatism Association (now the American College of Rheumatology), and it is currently a member of the spondyloarthropathy spectrum [2]. PsA was initially defined by Moll and Wright as ’an inflammatory arthritis in the presence of psoriasis with a usual absence of rheumatoid factor’. Since 2006, the CASPAR (classification criteria for psoriatic arthritis) has been used in order to diagnose PsA and differentiate it from other spondyloarthropathies (SpA). PsA is mainly characterized by stiffness, pain, swelling, and tenderness of the joints and their surrounding ligaments and tendons (dactylitis and enthesitis). The course of this disease is variable and unpredictable, ranging from mild and nondestructive joint involvement to a severe, debilitating, and erosive arthropathy [3]. The prevalence of PsA varies according to ethnicity, from 22.7% in European patients with psoriasis to 14.0% in Asian patients. Additionally, it has been found to be significantly lower in children and adolescents than in adults but equally frequent in both sexes [4].

The etiologic events that cause the development of PsA are complex and likely to be closely related to the mechanisms that underlie psoriasis [5]. Joint and bone changes have particular features of both spondylarthritis and rheumatoid arthritis (RA), as well. Available evidence indicates that these disorders show great complexity and heterogeneity and that genetic and environmental factors converge to trigger inflammatory events in multiple immune pathways [6]. Thus, understanding individual pathomechanistic factors of PsA and their interplay is essential for the identification of novel disease biomarkers and the clinical implementation of new individualized, target-directed, effective, and tolerable treatments [7]. Actually, the pharmacological treatment of PsA has evolved significantly over the last 15 years, thanks to the emergence of biologic therapies such as TNF inhibitors (TNFi), IL-17 inhibitors (IL-17i), IL-12/23 inhibitor (IL-12/23i), IL-23 inhibitors (IL-23i) and new targeted oral agents, in addition to the conventional disease-modifying antirheumatic drugs (DMARDs) [8,9] (Ogdie et al., 2020 guidelines) (Ianonne et al., 2020 PLoSone). The growing experience with these medications has revolutionized the approach to disease management in PsA.

The objective of this review is to bring together what has been described so far from genomics of PsA in order to encourage further research in this disease and other related pathologies such as ankylosing spondylitis (AS) to start creating evidence accordingly.

## 2. Genetic Factors of Susceptibility to Psoriatic Arthritis

The biological mechanisms for the development of PsA are not completely understood; however, there is strong evidence from family studies that support a relevant genetic component [10]. Karason et al. demonstrated that the effect of genetic factors in PsA extends over several generations, showing a 40-fold increase in the risk of developing PsA for first-degree relatives of individuals with PsA and decreasing this risk as the relationship is farther [11]. This and other observations indicate that PsA is a complex disease where interaction of multiple factors, environmental and random events, which occurs in genetically predisposed individuals, triggers pathological pathways that lead to the development of the disease [12]. In this sense, different approaches are being carried out in order to elucidate the genetic predisposition and biological mechanisms that give rise to PsA.

### 2.1. Human Leukocyte Antigen Region

The human major histocompatibility complex (MHC) region, human leukocyte antigen (HLA), consists of genes encoding key components of the immune system [13] and is divided into three subregions referred to as HLA classes I, II, and III. Despite there being a broader number of HLA class I genes, there are only three major ones that have been studied in detail: HLA-A, HLA-B, and HLA-C. For its part, the major HLA class II antigens are DR, DQ, and DP encoded by the gene pairs DRA and DRB, DQA and DQB, and DPA and DPB, respectively. During the development of the immune system, HLA molecules are binding to self-peptides selecting an individual’s specific T cell repertoire. HLA class I molecules present peptides derived predominantly from intracellular pathogens to CD8+ T cells, whereas HLA class II molecules present peptides from membrane and extracellular proteins to CD4+ T cells. In order to protect the host from pathogens and promote reproductive success, the number of HLA alleles has increased hugely. It has been described that certain allotypes exhibit the ability to bind self-peptides contributing to the development of different autoimmune diseases [14].

As occurs in other immune-mediated inflammatory diseases, the strongest genetic signal of susceptibility to PsA comes from the MHC region [15]. Psoriasis is strongly associated with the presence of HLA-Cw6, having been associated with different aspects of psoriasis such as genetic susceptibility, different clinical manifestations, or treatment efficacy [16]. Through candidate gene studies, five different HLA alleles have been associated with susceptibility to PsA: HLA-C*06:02, B*08:01, B*27:05, B*38:01, and B*39:01 [17]. In addition to this heterogeneity within the HLA associations, certain allotypes have been found associated with particular clinical features. The presence of the HLA-C*06:02 allele in PsA patients appears to be related to skin disease, whereas PsA is genetically more heterogeneous in patients who lack the classic psoriasis susceptibility gene HLA-Cw6 [17]. Different HLA-B alleles have been found significantly increased in PsA but not in the psoriasis cohort (B*08:01, B*27:05, B*38, and HB*39). Indeed, the frequency of the B*08:01 allele was found significantly decreased in the psoriasis cohort, suggesting a protective role for the development of psoriasis [17].

Furthermore, HLA-B*40:01 and HLA-B*44:01 alleles were found significantly decreased in PsA patients, suggesting that they might be protective against the development of the PsA. Okada et al. conducted a large study to characterize the genetic architecture of the HLA region in psoriasis patients, including 9247 affected individuals and 13,589 healthy controls of European ancestry, and imputing HLA class I and II genes from SNP genotype data [18]. Authors reported multiple HLA-C∗06:02-independent risk variants in both types of HLA genes for psoriasis susceptibility (HLA-A, -B and -C, and HLA-DQα1; as well as specific HLA alleles and amino acid positions). They also described that the amino acid at position 45 in the HLA-B molecule could lead to the risk heterogeneity between PsA and cutaneous psoriasis, showing that the presence of glutamate at that position increased PsA susceptibility. These data support previous studies where the presence of glutamate amino acid at position 45 was observed in different HLA-B alleles (B*08, B*27:05, B*38:01, and B*39:01) associated with PsA. Possible involvement of these alleles in PsA may be related to a common antigen presentation. Subsequently, Bowes et al. attempted to validate these previously reported associations within the HLA region to clarify the differentiation between patients with PsA and patients with cutaneous psoriasis alone, providing interesting results. They showed that the HLA-C*06:02 allele was not associated with PsA, whereas amino acid at position 97 of HLA-B was described as the most important risk factor for PsA development [19]. Curiously, that same variant represents the major AS risk factor [20].

Although a better understanding of the relevance of HLA alleles in disease phenotype is needed, it seems that there is great heterogeneity in the association between HLA alleles and PsA development, and this diversity also coexists with diverse genetic factors underlying overall psoriasis risk.

### 2.2. Non-HLA Genetic Associations by Genome-Wide Association Studies

In the last 15 years, our understanding of genetic susceptibility to complex disease has significantly improved, mainly thanks to the ever-increasing statistical power of genome-wide association studies (GWAS) [21]. GWAS is a wide-open analysis not restricted to regions of the chromosomes with previously described genes of interest. This hypotheses-free approach is considered to be a primary tool to identify inherited genetic variants associated with risk or protection against different diseases or particular traits. In these studies, hundreds of thousands, and even millions, of single nucleotide polymorphisms (SNPs) are genotyped with an unbiased view of the whole genome. Using GWAS, thousands of variants have been associated with hundreds of phenotypes, furthering our understanding of the genetic architecture of complex diseases, including SpA [22].

As well as the association between HLA and PsA, there is evidence for the involvement of other non-HLA genes in the pathogenesis of the disease [23]. The first GWAS performed to identify genetic factors predisposing to PsA was carried out in 2008 [24]. The sample size was limited to 667 PsA patients, but the authors were able to show the relevant role of HLA in the disease. Additionally, they also confirmed previously reported associations between interleukin 23 receptor (*IL23R*) and interleukin 12B (*IL12B*) with PsA. Subsequently, Hüffmeier et al. established IL12B as PsA susceptibility gene and identified new association with the TRAF3 (tumor necrosis factor receptor-associated factor 3) and interacting protein 2 gene (*TRAF3IP2*) [25], whereas Ellinghaus et al. reported that REL was an important candidate susceptibility locus for PsA [26]. Both studies were performed using independent validation groups. Already by 2015, Stuart et al. carried out the largest GWAS to date [27], including 1430 PsA cases and 1417 unaffected controls, and combining their data with previously reported genotyping studies, for a total of 3061 PsA cases, 3110 cutaneous psoriasis cases, and 13,670 unaffected controls. They detected 10 genomic regions associated with PsA, providing information about the pathogenetic similarities and differences between PsA and cutaneous psoriasis. At the same time, Bowes et al. also identify key insights into the genetics of PsA and fundamental differences between psoriasis and PsA [28]. Here, an immunochip genotyping array that contains ~196,000 SNPs covering immunogenetics genes related to the major autoimmune diseases was used to identify novel PsA susceptibility loci [29]. The authors confirmed the association for previously reported psoriasis susceptibility loci (*TRAF3IP2*, *IL12B*, *IL23R*, *IL23A*, *STAT2*, *TNIP1*, *TYK2*; among others), identified new PsA-specific risk loci at chromosome 5q31 and 1q31 (*SLC22A5* and *DENND1B* genes, respectively), and supported CD8+ T cells as a relevant cell type in the pathogenesis of the disease. More recently, Aterido et al. identified a new association between SNPs at the B3GNT2 locus and PsA [30]. They also performed pathway analyses, confirming the relevance of the glycosaminoglycan (GAG) metabolism pathway in PsA development and revealing new indications for approved drugs targeting this pathway.

Other studies have reported a genetic influence of killer cell immunoglobulin-like receptors (KIR) genes in PsA [22]. The KIR gene cluster constitutes a diverse family regarding gene content and allelic polymorphism, being an important component of natural killer (NK) cell target recognition, mainly through HLA class I molecules. Due to the small number of cases included in most studies, the conclusions of these studies must be taken with caution. The most robust results were those obtained by Chandran V et al. [31], where the frequency of KIR2DS2 was increased in PsA patients compared with healthy controls. It will be necessary to perform new studies analyzing the association between KIR genes and PsA development. HLA and KIR imputations based on SNP genotyping data will be important in this context [32].

In summary, the results from GWAS and successive meta-analyses have identified around 30 susceptibility non-HLA genes associated with PsA. We include a list of reported genome-wide PsA loci in Table 1 from the following studies [23,24,25,26,27,28,30,31,33]. Some of these genes have been described to be shared with other related diseases, such as AS, inflammatory bowel disease (IBD), RA or systemic lupus erythematosus (SLE) (Figure 1), being the most prominent genes: *ERAP1*, *ERAP2*, *IL23A*, *IL23R*, *REL*, *RUNX3*, *STAT4*, *TLR4*, *TNIP1*, *TRAF3IP2*, *TYK2*. The great majority of these genes can be classified in terms of their effect in the pathogenesis of PsA by activation or inhibition of the IL23/Th17 and nuclear factor kappa-light-chain-enhancer of activated B cells (NF-κB) signaling pathways, or also involving Janus kinase (JAK)/signal transducers and activators of transcription (STAT) signaling pathway (Figure 1).

### 2.3. Implications and Challenges for Personalized Medicine

To date, most patients included in GWAS are psoriasis-only or psoriasis vulgaris patients instead of PsA patients. This reality could explain the absence of PsA-specific genes. Thus, there is a strong need to carry out more GWAS with PsA patients exclusively, not just for gene identification only, but also to develop polygenic risk scores (PRS), which is the most potential application from GWAS data. PRS are quantitative scores that consider the collective effect of individual SNP markers in a given trait and could contribute to early diagnosis or prediction of the likelihood of developing some common diseases through the construction of models of risk-stratification. Recently, Knevel et al. reported a set of PRS (algorithm termed G-PROB) [34], providing information on the probability of an individual with inflammatory arthritis having RA, SLE, SpA, gout, and also PsA. Previously, Patrick MT et al. reported that a genetic risk score based on 200 markers had the ability to distinguish between PsA and cutaneous-only psoriasis [35]. They also showed that the genetic differences between the two phenotypes were due to regulatory elements, evidencing the need to integrate other omics data (transcriptomics and epigenomics inter alia), to make the implementation of personalized medicine to become a reality. While further research is required to validate these findings, these data offer a first approach to perform diagnostic/prognostic genetic risk profiling.

Pharmacogenetics, the use of genetic markers to predict response or toxicity to certain drugs, also has a huge potential mainly due to the relatively low cost and robustness of determinations. However, most studies focus on individual genetic markers, whereas their nature might be polygenic. Thus, the evaluation of genetic variants located within tumor necrosis factor-alpha (TNF-α) and their influence on the response to anti-TNFα have reported conflicting data in PsA [36]. Similarly, other genes involved in candidate signaling cascades, such as IL23/Th17 signaling pathway, could influence response to anti-IL17A and/or anti-IL-12/IL-23 therapies [36]. In any case, pharmacogenetic studies in PsA need to have an adequate sample size, including additional and integrative analysis with expression profiling.

## 3. Deriving Insights into Disease Biology through Multi-Omics Data Integration

The unstoppable advance of genomic medicine and the way it has changed the clinical management of many diseases is irrefutable. Genomic medicine has burst into clinical practice making it achievable to study the specific characteristics of some diseases in each patient and, among other things, to treat those patients with specific drugs, if available. This apparently simple but decisive change is the basis of precision medicine. However, for precision medicine to become a reality in complex diseases, such as PsA, it is necessary to take a step forward and include data obtained from other technologies that provide additional information about their biological characteristics, as well as having the capacity to integrate all the data generated.

### 3.1. Epigenetics

Epigenetic modifications can induce changes in chromatin without modifying the DNA sequence [37]. Mechanisms of epigenetics include DNA methylation/demethylation, histone modifications, and non-coding RNAs, such as microRNAs [38]. In addition, non-genetic factors, for example, diet, smoking, alcohol consumption, UV radiation, etc., can induce epigenetic changes that modify gene expression and thus play an important role in the development of diseases [39]. In particular, epigenetic modifications are known to play a major role in the pathophysiology of several immune-mediated disorders, such as psoriasis and PsA [40,41].

Studies in genetically identical monozygotic (MZ) twins with discordance in the clinical presentation of PsA have demonstrated the importance of epigenetics in this disease [42]. Likewise, parent-of-origin effects have been identified in humans in epidemiological analyses of multiple complex diseases, such as Ps and PsA. Different studies with large cohorts of PsA patients have repeatedly shown that there is a higher prevalence of psoriasis among offspring of psoriatic fathers compared to psoriatic mothers [43,44,45]. A significantly greater tendency of PsA probands to report an affected father compared to an affected mother has also been observed. In addition, these studies have also evidenced a significant reduction in the age of psoriasis onset in paternally transmitted cases, as well as an increase in the severity of PsA manifestations in subsequent generations [43,44,45]. These observations suggest that epigenetic modifications may contribute to the lack of heritability in Ps and PsA [46,47].

To date, there is not a large number of studies investigating DNA methylation/demethylation or histone modifications in PsA (Table 2). Most published studies focus on trying to understand the role of epigenetics in the observed differences in paternal versus maternal transmission and between drug responder/non-responder patients [48].

#### 3.1.1. DNA Methylation

In 1996, Kim et al. published the first study of DNA methylation in PsA patients demonstrating that peripheral blood mononuclear cells (PBMCs) from PsA patients present a specific methylation signature [48]. Genome-wide DNA methylation of PBMCs was lower in patients with PsA compared to healthy controls. This suggests that PsA is associated with DNA hypomethylation. This finding is in line with results observed in other autoimmune diseases, such as psoriasis, RA, and SLE that also showed reduced global DNA methylation in T cells and monocytes compared to healthy controls [49,50]. In all these studies, DNA hypomethylation was associated with reduced expression of the DNMT1 gene [51]. This gene encodes the enzyme DNA methyltransferase 1 that transfers methyl groups to cytosine nucleotides of genomic DNA. This protein is the major enzyme responsible for maintaining methylation patterns following DNA replication.

**Table 2 jpm-12-00035-t002:** Epigenomic studies in psoriatic arthritis.

Epigenetic Mechanism	Sample	Molecule	Variation	References
DNA methylation	PBMCs	Specific methylation signature	Hypomethylation	[48]
	Sperm cells	DYSFIP1, ADARB2, MBP, PRKAG2, ITGB2, OSBPL5, TNS3, and SNORD115	Hypermethylation	[52]
		HCG26, H19, and MIR675	Hypomethylation	
	Blood	MICA, IRIF1, PSORS1C3, and TNFSF4	Hypermethylation	[53]
		PSORS1C1C1	Hypomethylation	
Histone modifications	PBMCs	H4	Hypoacetylation	[54]
	HDMEC	H3, and H4	Hyperacetylation	[55]
Regulation via non-coding RNAs	Circulating CD14+ monocytes	miRNA-146a and miRNA-941	Upregulated	[56,57]
	PBCs	hsa-miR-126-3p, hsa-miR-151a-5p, hsa-miR-130a-3p, hsa-miR-199a-3p and hsa-miR-451a	Downregulated	[58] *
		hsa-miR-4741, hsa-miR-3196, hsa-miR-575, hsa-miR-3135b and hsa-miR-574-5p	Upregulated	

* Included the five most up/downregulated miRNAs. For a complete list, please read Pelosi et al., 2018 [58]. Abbreviations: HDMEC, human dermal microvascular endothelial cell; PBCs, peripheral blood cells; PBMCs, peripheral blood mononuclear cells.

Recently, Pollock et al. have identified several differentially methylated regions (DMRs) in sperm cells from PsA patients. They found changes in the methylation levels of genes associated with skin and/or joint disease. DYSFIP1, ADARB2, MBP, PRKAG2, ITGB2, OSBPL5, TNS3, and SNORD115 were hypermethylated, whereas HCG26, H19, and MIR675 genes were found hypomethylated [52]. In an interesting study, O’Rilley et al. demonstrated that the global DNA methylation pattern in paternally transmitted PsA differs from maternally transmitted PsA [53]. The authors analyzed whole blood samples in a cohort of 24 PsA patients and 24 healthy controls and found that the genes MICA, IRIF1, PSORS1C3, and TNFSF4 were hypermethylated, whereas PSORS1C1C1 was hypomethylated in paternally compared to maternally transmitted disease.

#### 3.1.2. Histone Modifications

Post-translational modifications of N-terminal amino acid residues within histone proteins, such as, for example, their acetylation and/or methylation, result in changes in their electrical charge and thus regulate their three-dimensional composition. Through this mechanism, histone modifications can mediate transcriptional activation or repression based on the location and nature of the post-translational modifications [59,60]. Ovejero-Benito et al. found decreased histone H4 acetylation in PBMCs from PsA patients compared to healthy controls [54]. A family of histone deacetylases, the sirtuins (SIRT1-SIRT7), have been implicated in the pathogenesis of PsA. Their main function is to suppress gene transcription by epigenetic remodeling [61]. Hu et al., 2018 found increased levels of anti-Sirt1 autoantibodies in the serum of PsA patients, but not in RA patients or healthy controls [62]. Likewise, inhibition of sirtuin-1 results in increased acetylation of histones H3 and H4 and decreased secretion of inflammatory chemokines (CXCL10, CCL2, and CXCL8). This alters inflammatory responses to TNF-a and IL-1b, which are part of the inflammatory cascades of PsA [55]. From the results of this study, it can be concluded that the likely involvement of SIRT1 in the pathogenesis of PsA demonstrates how epigenetic mechanisms are closely related to inflammation and tissue damage.

#### 3.1.3. Transcriptional Regulation via Non-Coding RNAs

Non-coding RNAs (ncRNAs) are RNA molecules without protein translation potential but capable of modulating gene expression through different mechanisms [63]. NcRNA-mediated gene silencing constitutes an important type of epigenetic alterations, and ncRNAs are identified as playing important roles in normal physiologic processes, complex human traits, and human diseases [64,65,66]. ncRNAs signatures in PsA have been primarily linked to T cell activation and cytokine signaling, cell proliferation and inflammation, and cartilage/bone metabolism [56,67,68].

Several studies have demonstrated differential expression of miRNAs that had previously been found to be altered in blood and synovial fluid samples from RA patients. For example, miRNA-146a and miRNA-941 expression is known to be upregulated in circulating CD14+ monocytes from PsA patients [56,57]. Pelosi et al. identified in blood cells specific miRNA signatures associated with active and non-active disease. These miRNAs target pathways relevant in PsA, such as TNF, MAPK, and WNT signaling cascades. miR-126-3p was the most downregulated miRNA in active PsA patients [58].

Globally, the current understanding of the role of epigenetics in PsA is very limited due to the few studies available. In the future, new studies will shed light on this role and improve the understanding of the factors involved in its development.

### 3.2. Transcriptomics

The transcriptome is the complete set of transcripts expressed by a cell or tissue at a given time or under specific physiological conditions [69]. Transcriptome analysis is essential to understand the molecular mechanisms involved in specific biological processes and in the onset and development of diseases based on information about gene structure and function. Moreover, transcriptome analysis delivers an unprecedented level of detail in understanding cellular phenotypes by examining genes expressed in specific physiological and pathological states.

#### 3.2.1. Transcriptomics in Peripheral Blood

Blood is the most important biological sample. It contains different types of RNAs, proteins, and metabolites secreted by the cells that can provide valuable information about pathological processes. In PsA, the most used samples in human studies are peripheral blood and tissue-derivates (Table 3). Batliwalla et al. studied PBCs in a cohort consisting of 19 PsA patients and 19 sex- and age-matched controls. Increased gene expression was observed in three proinflammatory genes (S100A8, S100A12, and TXN). The most downregulated expression was observed in several genes involved in downregulation or suppression of innate and acquired immune responses (SIGIRR, STAT3, SHP1, IKBKB, IL-11RA, and TCF7) [70]. Both findings suggest an inappropriate control of the immune response in PsA patients that favor a proinflammatory state. Stoeckman et al. analyzed gene expression profiles in peripheral blood samples from PsA patients. Interestingly, they did not find a large overlap with the expression profiles of patients with RA, despite both being systemic inflammatory disorders. Only three genes (RAB13, RAB32, and FCGBP) were coordinately and differentially expressed in both diseases. On the other hand, four genes (ZNF395, DDX28, PCNX3, and PI3KC2B) showed the highest discriminatory potential, suggesting that they may be useful in confirming the clinical diagnosis of PsA [71]. The author also used public databases for collecting gene expression data of SLE patients. The differences observed in the gene expression profile in PsA patients compared with RA and SLE profiles was a clear indicator that different pathophysiological mechanisms operate in these diseases [71].

#### 3.2.2. Transcriptomics in Tissues

Due to the importance of inflammatory components in PsA, the analysis of affected tissues may offer additional and valuable insights. In this sense, Dolcino et al. used PBCs and synovial biopsies of 10 patients with PsA to study gene expression. Importantly, these patients have not been previously treated with anti-TNF agents or with disease-modifying antirheumatic drugs (DMARDs). Among the differentially expressed genes, authors observed the upregulation of Th17 related genes and of type I interferon (IFN) inducible genes [72]. The synovial transcriptome showed gene clusters involved in pathogenic processes of PsA, such as inflammation, angiogenesis, and bone remodeling [72]. The authors also identified a novel protein called osteoactivin as a possible disease biomarker.

In a similar approach, Belasco et al. collected paired PsA synovial tissue and skin samples from 12 patients to study representative genes of the inflammatory process. Many upregulated genes in the skin, such as S100A7, S100A8, and S100A9, were IL-17 signature genes [73]. In synovium, the top differentially expressed genes were related to cartilage and bone breakdown and formation (MMP1, COL2A1, WISP1, HAS1, IBSP, FZD8, BMPR2, and WNT3A) or the angiogenesis that is present in PsA (COL18A1, F5, VEGF, and TGFB1) [73].

#### 3.2.3. Single-Cell mRNA Sequencing

Single-cell mRNA sequencing (scRNA-seq) is a novel technology that permits direct comparison of the transcriptomes of individual cells from the same biological sample. Therefore, the main use of scRNA-seq in the last years has been to assess transcriptional similarities and differences within a population of cells. In some cases, this new approach has permitted the discovery of cellular heterogeneity that had never been seen before, for example, immune cells [75,76]. In a scRNA-seq study conducted by Penkava et al., this transcriptomic technology revealed CD8 T cell clonal expansions within the joints expressing cycling, activation, tissue-homing, and tissue residency markers. Results showed that CXCR3, CXCR6, CCR5, and CCR2 are highly expressed on synovial T cells, and genes CXCL9 and CXCL10 are also highly expressed in the synovial fluids [74].

### 3.3. Machine Learning-Based Analysis of Multi-Omics Data

High-throughput technologies applied to health sciences provide us with priceless information on the molecular characteristics of diseases, and this information is obtained through the processing of a huge amount of data. Managing this massive amount of data is a challenge for the scientific community. In recent years, bioinformaticians have become key players in many laboratories and hospitals. Nowadays, genomic data generated by next-generation sequencing (NGS) technologies are already being used for disease diagnosis. For example, disease-specific gene panels are widely used in clinical practice. In cases where the genes analyzed do not allow a satisfactory diagnosis to be reached, clinical exome analysis can be a great alternative. Partly, this progress is a reality due to the existence of big private companies (Illumina, Thermo Fisher, etc.), which invest huge amounts of money in research and development, allowing a larger number of genetic diseases to be diagnosed every day.

Despite this breakthrough, it is widely known that most genetic diseases are, in fact, complex diseases and are caused by a combination of genetic, environmental, and lifestyle factors [77]. In these diseases, the analysis of genetic variants separately is not enough. As explained above, epigenetics studies the alteration of gene expression without modification of the DNA nucleotide sequence, and thus, it becomes essential to understand the molecular basis of diseases. On the other hand, transcriptomics is proving crucial for understanding the behavior of many diseases since this technology allows us, for example, to know the changes in gene expression that occur as a result of various factors, such as drug treatment. Therefore, there is no doubt that it will be necessary to combine the information provided by these and other -omics technologies if we want to achieve a true understanding of the molecular basis of diseases [78]. Multi-omics data generated for the same set of samples can provide useful insights at multiple levels, and so the development of algorithms that permit their integration will be key [78]. The access of epigenetics and transcriptomics to clinical practice will be linked to our ability to integrate and understand the data obtained from them.

In this context, machine learning is a new concept in the healthcare ecosystem that has recently emerged and will be crucial for the development of precision medicine. Machine learning can be defined as a discipline in the field of artificial intelligence (AI), which, through algorithms, gives computers the ability to identify patterns in massive data and make predictions [79]. The quality of the databases used to train these algorithms is therefore fundamental, as the accuracy of the results obtained will depend on it [79]. From the point of view of medicine, as high-throughput technologies advance and provide us with a larger amount of data, machine learning algorithms will become more and more important. Although machine learning is currently in its very early stages, various approaches are already being developed to study several diseases such as cancer, heart diseases, dermatological diseases, diabetes, and neurological disorders [80]. Apart from detecting and predicting the evolution of diseases, machine learning is also being used for other purposes that can have a major impact on improving the clinical management of patients in the future. The structuration of medical information contained in health records or the conversion of information contained in medical images into structured information are two interesting examples [81].

Finally, those diseases that are currently difficult to diagnose and manage will undoubtedly benefit the most from the revolution resulting from high-performance technologies, multi-omics data integration, and the use of machine learning algorithms. In the case of PsA, it should be possible to predict or detect it at earlier stages, as well as to obtain a better characterization against other disorders with common features, such as Ps and RA. These advances should make it feasible to cluster patients according to their molecular profiles, applying targeted and more effective treatments that increase the likelihood of the patient being cured.

## 4. Conclusions

Studies carried out to date in the field of genomics have provided interesting information on PsA. However, despite the number of genes identified, nowadays, it is only possible to partially elucidate its genetic basis. Possibly, its greater clinical heterogeneity, compared to other autoimmune diseases, such as Ps or RA, together with the smaller number of correctly diagnosed patients, have influenced the lack of significant and reproducible genetic and genomic findings. Although these studies have advanced our understanding of its pathogenesis, the exact pathophysiological mechanisms underlying PsA expression remain incompletely understood and need further investigation. In this context, there is a strong need for genome-wide association studies on patients with PsA, including PsA-weighted or specific variants, as well as for a better understanding of the relevance of HLA alleles in disease expression. An effort is also required to increase the ability to detect the genetic variants that create a predisposition to psoriatic disease and to predict response to biological therapy.

As explained in detail in this review, the additional use of other high-throughput omics technologies, particularly epigenomics and transcriptomics, enables to study of diseases at different levels and obtain valuable information. The integration of all this omics data and the use of machine learning algorithms undoubtedly promises that in the coming years, we will find ourselves in an ideal situation to better understand the molecular basis of complex diseases, such as PsA. This great achievement will certainly allow the decisive development of precision medicine in clinical practice.

## Figures and Tables

**Figure 1 jpm-12-00035-f001:**
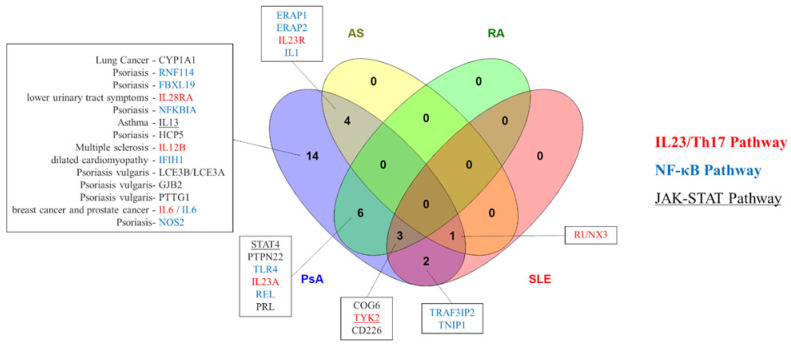
Shared non-human leukocyte antigen susceptibility genes between psoriatic arthritis and other diseases. Abbreviations: AS, ankylosing spondylitis; IBD, inflammatory bowel disease; PsA, psoriatic arthritis; RA, rheumatoid arthritis; SLE, systemic lupus erythematosus.

**Table 1 jpm-12-00035-t001:** Genome-wide significant genetic variations associated with psoriatic arthritis.

Genes	Chromosome	Population
*HLA class I molecules*		
**HLA-B*08**	6	EUR
**HLA-B*27**	6	EUR
**HLA-B*38**	6	EUR
**HLA-B*39**	6	EUR
*Non-HLA genes*		
** *IL23R* **	1	EUR
** *PTPN22* **	1	EUR
*RUNX3*	1	EUR
*IL28RA*	1	EUR, EAS
*LCE3B/LCE3A*	1	EAS
*REL*	2	EUR
*IFIH1*	2	EUR
*STAT4*	2	EUR
*IL1*	2	EUR
*B3GNT2*	2	EUR
** *CSF2/P4HA2* **	5	EUR
*TNIP1*	5	EUR, EAS
*IL12B*	5	EUR, EAS
*ERAP1*	5	EUR, EAS
*ERAP2*	5	EUR
*PTTG1*	5	EAS
*TNFA*	6	EUR
*HCP5*	6	EAS
** *TNFAIP3* **	6	EUR
*TRAF3IP2*	6	EUR
*TLR4*	9	EUR
*IL23A*	12	EUR
*GJB2*	13	EAS
*COG6*	14	EUR
*NFKBIA*	14	EUR
*FBXL19*	16	EUR
*NOS2*	17	EUR
*CD226*	18	EUR
*TYK2*	19	EUR
*KIR2DS2*	19	EUR
*RNF114*	20	EUR

Bold indicates specific to psoriatic arthritis susceptibility. Population indicates the ancestral diversity of associations according to the GWAS catalog (EAS: East Asia, EUR: Europe).

**Table 3 jpm-12-00035-t003:** Transcriptomic studies in psoriatic arthritis.

Origin	Sample	Genes	Deregulation	References
Peripheral blood	PBCs	S100A8, S100A12 and TXN	Upregulated	[70]
		SIGIRR, STAT3, SHP1, IKBKB, IL-11RA, and TCF7	Downregulated	
	Blood	RAB13, RAB32, and FCGBP	Upregulated	[71]
		ZNF395, DDX28, PCNX3, and PI3KC2B	Downregulated	
Tissues	PBCs and synovial biopsies	Th17 related genes, IFN inducible genes	Upregulated	[72]
	Skin	S100A7, S100A8, and S100A9	Upregulated	[73]
	Sinovium	MMP1, COL2A1, WISP1, HAS1, IBSP, FZD8, BMPR2, and WNT3A	Upregulated	
	Synovial T cells	CXCR3, CXCR6, CCR5, and CCR2	Upregulated	[74] *
	Synovial fluid	CXCL9 and CXCL10	Upregulated	

* Single-cell mRNA sequencing. Abbreviations: PBCs, peripheral blood cells.

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
