# Peer review of "Advances of Genomic Medicine in Psoriatic Arthritis"

_jpm, 2022, doi:10.3390/jpm12010035_

Round 1

Reviewer 1 Report

Comprehensive and very well written!

In part: Histone modifications: Ovejero- Benito and all found histone...from PsA patients, but in original reference it is mentioned in psoriasis patients (Ref.52)-please comment!

Author Response

  1. In part: Histone modifications: Ovejero- Benito and all found histone...from PsA patients, but in original reference it is mentioned in psoriasis patients (Ref.52)-please comment!

R: Thank you for the comment. Ovejero-Benito et al. found decreased histone H4 acetylation in PBMCs from PsA patients compared to healthy controls, not H3. The error has been corrected in the main text and Table 2.

Reviewer 2 Report

A very interesting narrative review focusing on the relationship between psoriatic arthritis and genomic. I really enjoyed reading it . There are some comments that needs to be addressed before publication:

English language needs some revisions for example, Page 2 line 1 It seems that arthropathic psoriasis is present in 14% of Asian patients....please correct.

A small subchapter about current treatments for psoriasis  and psoriatic arthritis should be added in the introduction; here some articles you could embody: doi: 10.1016/S0140-6736(18)30949-8. doi: 10.1371/journal.pone.0241575. doi: 10.1111/dth.14504. doi: 10.1093/rheumatology/kez383.

Good Luck

Author Response

  1. English language needs some revisions for example, Page 2 line 1 It seems that arthropathic psoriasis is present in 14% of Asian patients....please correct.

R: The manuscript has been deeply revised and this particular sentence has been modified in accordance.

  1. A small subchapter about current treatments for psoriasis and psoriatic arthritis should be added in the introduction; here some articles you could embody: doi: 10.1016/S0140-6736(18)30949-8. doi: 10.1371/journal.pone.0241575. doi: 10.1111/dth.14504. doi: 10.1093/rheumatology/kez383.

R: Following the reviewer comments, the next paragraph indicating the current treatments for psoriasis and psoriatic arthritis has been included as a small subchapter in the Introduction (page 2):

“The treatment of PsA has evolved significantly over the last 15 years, thanks to the emergence of biologic therapies such as TNF inhibitors (TNFi), IL-17 inhibitors (IL-17i), IL-12/23 inhibitor (IL-12/23i), IL-23 inhibitors (IL-23i) and new targeted oral agents, in addition to the conventional disease modifying antirheumatic drugs (DMARDs) [8,9] (Ogdie et al., 2020 guidelines) (Ianonne et al., 2020 PLoSone). The growing experience with these medications has revolutionized the approach to disease management in PsA.”